# Integration of Metabolomes and Transcriptomes Provides Insights into Morphogenesis and Maturation in *Morchella sextelata*

**DOI:** 10.3390/jof9121143

**Published:** 2023-11-27

**Authors:** Chen Zhang, Xiaofei Shi, Jiexiong Zhang, Yesheng Zhang, Wei Liu, Wen Wang

**Affiliations:** 1School of Ecology and Environment, Northwestern Polytechnical University, Xi’an 710072, China; zhangchen1996@mail.nwpu.edu.cn (C.Z.); zhangjx_light@mail.nwpu.edu.cn (J.Z.); 2The Germplasm Bank of Wild Species, Yunnan Key Laboratory for Fungal Diversity and Green Development, Kunming Institute of Botany, Chinese Academy of Sciences, Kunming 650201, China; shixiaofei@mail.kib.ac.cn; 3Shandong Junsheng Biotechnologies Co., Ltd., Liaocheng 252400, China; zhangyesheng1987@163.com

**Keywords:** morphogenesis, ascocarp maturation, *Morchella sextelata*, metabolome, transcriptome

## Abstract

True morels (*Morchella*, Pezizales) are a popular edible and medicinal fungus with great nutritional and economic value. The dynamics and regulatory mechanisms during the morphogenesis and maturation of morels are poorly understood. In this study, the metabolomes and transcriptomes of the mycelium (MY), primordium differentiation (PR), young fruiting body (YFB), and mature fruiting body (MFB) were comprehensively analyzed to reveal the mechanism of the morphogenesis and maturation of *Morchella sextelata*. A total of 748 differentially expressed metabolites (DEMs) and 5342 differentially expressed genes (DEGs) were detected, mainly enriched in the carbohydrate, amino acid, and lipid metabolism pathways, with the transition from the mycelium to the primordium being the most drastic stage at both the metabolic and transcriptional levels. The integrated metabolomics and transcriptomics highlighted significant correlations between the DEMs and DEGs, and specific amino acid and nucleotide metabolic pathways were significantly co-enriched, which may play key roles in morphological development and ascocarp maturation. A conceptual model of transcriptional and metabolic regulation was proposed during morphogenesis and maturation in *M. sextelata* for the first time, in which environmental factors activate the regulation of transcription factors, which then promote metabolic and transcriptional regulation from vegetative to reproductive growth. These results provide insights into the metabolic dynamics and transcriptional regulation during the morphogenesis and maturation of morels and valuable resources for future breeding enhancement and sustainable artificial cultivation.

## 1. Introduction

True morels (*Morchella*, Pezizales) are edible mushrooms highly prized for their special flavor as well as their nutritional and economic value [1]. With unique honeycomb-like pileus tissues, there has been a global cultural tradition of consuming morel ascocarps [2,3]. Numerous studies have shown morels are rich in amino acids, polysaccharides, and trace elements [1,4,5], and have anti-inflammatory, antioxidant and immune-enhancing properties [6,7]. Due to their valuable edible and medicinal qualities and the increasing global markets, artificial cultivation of morels has been continuously explored, helping to satisfy widespread global market demand [8].

True morels generally represent species of the genus *Morchella*, as distinguished from the morphologically similar species of the genera *Helvella*, *Verpa,* and *Gyromitra* [8]. To date, the genus *Morchella* contains more than 78 separate phylogenetic species, consisting of three evolutionary clades: The Rufobrunnea clade, the Esculenta clade, and the Elata clade [9]. Several species have been reported to be successfully cultivated, such as *M. rufobrunnea* in the United States [10] and *M. importuna*, *M. sextelata,* and *M. eximia* in China [11]. In the last decade, the artificial cultivation of morels has increased rapidly in China, reaching an area of 16,466 ha in 2021–2022 [12]. Cultivation methods are mainly based on the domestication of easy-to-fruit varieties, the utilization of exogenous nutrient bags, and suitable environmental conditions [11,13]. Despite remarkable breakthroughs in artificial cultivation, approximately half of morel cultivations experience low fruiting rates and difficulties in obtaining stable profits [9,12], caused by serious problems such as strain aging or degradation [9,14], fungal and bacterial diseases [12,15,16,17], environmental fluctuations, and refined field management [11,13]. These problems place greater demands on understanding the developmental mechanism and biology of morels.

A generalized morel life cycle consists of the mycelium, primordium, and young and mature fruiting body; in brief, the vegetative mycelium growth completes the assimilation, absorption, and storage of nutrients, then turns to reproductive growth and develops into ascocarps under the stimulation of appropriate environmental conditions [8,9]. The special nutritional supply method using exogenous nutrient bags is key to the successful cultivation of morels. The sufficient carbon and nitrogen sources in the nutrient bags provide energy for the growth of the mycelium network in the soil, meeting the needs for later reproductive growth. Of course, a portion of the nitrogen source in the soil is also absorbed [13]. In addition, our and others’ studies have shown that primordium formation and fruiting body maturation are closely related to soil microbiota, which may promote morel growth [18,19,20,21,22]. Despite substantial processes in the morel life cycle [9,23], nutrient metabolism [13,19,24], and microbiota interactions [18,19,20,21,22], little is known about the molecular regulation mechanisms during morphogenesis and maturation in morels, especially the regulation mechanism from vegetative to reproductive growth, which has seriously hampered sustainable artificial cultivation.

To explore the transcriptional and metabolic regulation during the morphological development and ascocarp maturation of *M. sextelata*, we obtained and analyzed the dynamics of metabolomes and transcriptomes during the MY, PR, YFB, and MFB stages. In addition, integrated metabolomics and transcriptomics were conducted to investigate the correlation between differentially expressed metabolites (DEMs) and differentially expressed genes (DEGs), and a conceptual model for transcriptional and metabolic regulation during the morphogenesis and maturation of morels is proposed for the first time. The present integrated study effectively addresses the gap in knowledge of metabolomics and transcriptomics related to the morphogenesis and maturation of morels.

## 2. Materials and Methods

### 2.1. Sample Collection

An *M. sextelata* strain, referred to as D1, was grown at 20 °C in potato dextrose agar medium (containing 6 g/L of potato extract, 20 g/L of dextrose, and 15 g/L of agar) for 7 days, and the mycelial samples (MY) were collected and quickly frozen in liquid nitrogen and then stored in a −80 °C freezer. Fresh cultivated morels were collected from a farm (25.16° N, 99.24° E) in the Yunnan province of China during the fruiting stage. The fresh samples from PR, YFB, and MFB were collected and quickly frozen in liquid nitrogen and then stored in a –80 °C freezer. The features of the four sampling stages are shown in Figure 1a and described as follows: MY: white overall, thick and dense mycelia, well-developed aerial hyphae, and a few sclerotia formed; PR: 0.5–1 cm in size, smooth surface, white or light grey overall, and slightly swollen cylinder; YFB: 5–6 cm in size, surface with unfolded ridges, light yellow or light brown pileus, white stipe and rapid growth; MFB: over 10 cm in size, unique honeycomb-like dark brown or black pileus, white stipe, with ascospore formation.

### 2.2. Metabolite Extraction

A total of twenty-four samples were collected for metabolomics, with six biological replicates in each stage. The collected samples (100 mg) were individually grounded with liquid nitrogen and the homogenate was resuspended with prechilled 80% methanol by vortexing well. The samples were incubated on ice for 5 min and then centrifuged at 15,000× *g* and 4 °C for 20 min. The supernatant was diluted to a final concentration containing 53% methanol by LC-MS-grade water. The samples were subsequently transferred to a fresh Eppendorf tube and then centrifuged at 15,000× *g* and 4 °C for 20 min. Finally, the supernatant was injected into the LC-MS/MS system analysis as described previously [25].

### 2.3. UHPLC-MS/MS Analysis

UHPLC-MS/MS analyses were performed using a Vanquish UHPLC system (ThermoFisher, Waltham, MA, USA) coupled with an Orbitrap Q Exactive^TM^ HF mass spectrometer (ThermoFisher, Waltham, MA, USA) in Novogene Co., Ltd. (Beijing, China). Samples were injected onto a Hypersil Goldcolumn (100 × 2.1 mm, 1.9 μm) using a 17 min linear gradient at a flow rate of 0.2 mL/min. The eluents for the positive polarity mode were eluent A (0.1% FA in water) and eluent B (methanol). The eluents for the negative polarity mode were eluent A (5 mM ammonium acetate, pH 9.0) and eluent B (methanol). The solvent gradient was set as follows: 2% B, 1.5 min; 2–85% B, 3 min; 85–100% B, 10 min; 100–2% B, 10.1 min; 2% B, 12 min. The Q Exactive^TM^ HF mass spectrometer was operated in positive/negative polarity mode with a spray voltage of 3.5 kV, a capillary temperature of 320 °C, a sheath gas flow rate of 35 psi, an aux gas flow rate of 10 L/min, an S-lens RF level of 60, and an aux gas heater temperature of 350 °C.

### 2.4. Metabolite Identification and Quantification

The raw data files generated by UHPLC-MS/MS were processed using the Compound Discoverer 3.1 (ThermoFisher, Waltham, MA, USA) to perform peak alignment, peak picking, and quantitation for each metabolite. The main parameters were set as follows: retention time tolerance, 0.2 min; actual mass tolerance, 5 ppm; signal intensity tolerance, 30%; signal/noise ratio, 3; minimum intensity, etc., then peaks were matched with the mzCloud (https://www.mzcloud.org/ (accessed on 13 April 2021)), mzVault, and MassList databases to obtain accurate qualitative and relative quantitative results.

These metabolites were annotated using the KEGG database v98.0 (https://www.genome.jp/kegg/pathway.html (accessed on 13 April 2021)), HMDB database v4.0 (https://hmdb.ca/metabolites (accessed on 13 April 2021)), and LIPIDMaps database (http://www.lipidmaps.org/ (accessed on 13 April 2021)). Principal components analysis (PCA) and partial least squares discriminant analysis (PLS-DA) were performed using metaX (http://metax.genomics.cn/ (accessed on 13 April 2021)) [26]. The metabolites with a variable importance in projection (VIP) >1, a *p*-value < 0.05 and a |Log2(foldchange)| ≥ 1 were considered to be significantly differentially expressed. The K-means clustering of the differentially expressed metabolites was performed with R package Mfuzz v2.60.0 [27]. KEGG pathway enrichment of the differential metabolites was performed, which was considered statistically significant when the *p*-value was <0.05. Visualization of data was performed with R and TBtools [28].

### 2.5. RNA Extraction and Sequencing

Twelve samples were collected for transcriptomics, with three biological replicates in each stage. Total RNA was extracted using the TRIzol reagent (Invitrogen, Carlsbad, CA, USA), and their quality was assessed using the agarose gel electrophoresis and Agilent 2100 system (Agilent, Santa Clara, CA, USA) with RIN (RNA integrity number). Sequencing libraries were generated using a NEBNext Ultra RNA Library Prep Kit for Illumina (NEB, San Diego, CA, USA) following the manufacturer’s recommendations. The qualified libraries were pooled and sequenced on the Illumina Novaseq 6000 platform (Illumina, San Diego, CA, USA) with PE150 strategy in Novogene Bioinformatics Technology Co., Ltd. (Beijing, China). Raw sequencing reads were deposited in the National Center for Biotechnology Information (NCBI) database under accession number PRJNA989072.

### 2.6. Transcriptomic Analysis

Quality filtering was performed using fastp v0.23.2 with default parameters [29], and high-quality reads were then mapped to the reference genome (Genebank ID: GCA_024713665.1) using HISAT2 v2.2.1 [30]. The FPKM (Fragments Per Kilo base of transcript sequence per Millions of base pairs sequenced) of each gene was calculated on the basis of the length of the gene and the number of reads mapped to this gene. The R package DESeq2 v1.34.0 was used to identify the differentially expressed genes (DEGs) between the four sample sets [31]. The raw reads of counts ≥10 were used as the input in the DESeq2 analysis. The Wald test was used for testing differential gene expression, and the normal shrinkage estimator was used to adjust fold changes. The *p*-value was adjusted using Benjamini and Hochberg’s approach (Padj). Genes satisfying the criteria of a |Log2(foldchange)| ≥ 1 and a Padj < 0.05 were identified as DEGs. KEGG and GO annotations were performed with the eggNOG database [32], and enrichment analysis was performed using clusterProfiler v4.2.2 based on Fisher’s exact test [33], which was considered statistically significant when the adjusted *p*-value was <0.05 using the Benjamini and Hochberg’s approach. Transcription factors (TFs) were identified from the lists of significant DEGs via the pipeline of the Fungal Transcription Factor Database (FTFD) v1.2 (http://ftfd.snu.ac.kr/ (accessed on 16 October 2022)) and Plant Transcription Factor Database (PlantTFDB) v5.0 (http://planttfdb.cbi.pku.edu.cn/ (accessed on 16 October 2022))) [34,35]. Carbohydrate-active enzymes (CAZymes) were annotated via the dbCAN3 web server [36], which integrates three databases (CAZy database, dbCAN HMMdb, and dbCAN-sub HMMdb) for automated CAZyme annotation. Visualization of data was performed with R and TBtools [28].

### 2.7. Correlation Analysis between the Metabolome and Transcriptome Data

Pearson correlation coefficients (PCCs) were calculated for metabolome and transcriptome data integration. In this study, log conversion of the data was performed uniformly before analysis. For the joint analysis between the metabolomes and transcriptomes, the correlation was assessed by the *cor* function of the stats v4.1.0 R package by applying a 0.95 PCC threshold. A nine-quadrant plot was established using the MouseLittle tools (http://cloud.keyandaydayup.com/ (accessed on 20 March 2023)), a free online platform for data analysis.

### 2.8. Quantitative Real-Time PCR (qRT-PCR) Validation

RNA isolation and RT-PCR were performed as previously reported [37]. Total RNA was isolated using an RNAiso Plus Kit (Takara, Tokyo, Japan) and reverse-transcribed using a HiScript II One-Step RT-PCR Kit (Vazyme, Nanjing, China). qRT-PCR was performed using the CFX Connect Real-Time PCR System (Bio-Rad, Hercules, CA, USA). The 10 μL reaction mixture contained 0.5 μL of cDNA (15 ng), 0.5 μL of primers (10 μM), 5 μL of SYBR qPCR Master Mix (Vazyme, Nanjing, China) and 3.5 µL of ddH_2_O. Three biological and technical replicates were used. The CYC3 gene was selected as the internal reference gene, which has been evaluated as a stable internal reference gene in morels [38]. The primers used for qRT–PCR are shown in Appendix A. Relative gene expression was analyzed using the 2^−ΔΔCt^ method [39].

## 3. Results

### 3.1. Metabolomics Analysis during Morphogenesis and Maturation in M. sextelata

#### 3.1.1. Quality Control of the Metabolome Data

To understand metabolic dynamics during morphogenesis and maturation, non-targeted metabolomics (UHPLC-MS/MS) were detected for four representative developmental stages (Figure 1a), namely, the mycelium (MY), primordium differentiation (PR), young fruiting body (YFB), and mature fruiting body (MFB) stages. The correlation analysis of the QC samples showed highly stable and reliable metabolic data quality (Appendix A).

A total of 943 annotated metabolites were identified in *M. sextelata* (Appendix A), of which 661 were classified into 9 groups: 273 lipids and lipid-like molecules, 134 organic acids and derivatives, 87 organoheterocyclic compounds, 46 organic oxygen compounds, 45 benzenoids, 42 nucleosides, nucleotides and analogs, 21 phenylpropanoids and polyketides, 11 organic nitrogen compounds, and 2 alkaloids and derivatives (Figure 1b). PCA analysis showed that the samples within each stage clustered together, with four clusters clearly separated (Figure 1c). Partial least squares discriminant analysis (PLS-DA) showed that different developmental stages also showed separate clusters (Appendix A). These results indicate the different developmental stages of *M. sextelata* have different characteristic metabolic compositions.

#### 3.1.2. Identification and Screening of DEMs

A comparative analysis was performed to identify differences in the metabolite expression patterns. A VIP score ≥ 1, a |Log2(foldchange)| ≥ 1, and a *p*-value < 0.05 were considered as screening criteria for significant differences. The PLS-DA results showed that 425 (257 upregulated and 168 downregulated), 271 (171 upregulated and 100 downregulated), and 269 (116 upregulated and 153 downregulated) DEMs were identified in MY vs. PR, PR vs. YFB, and YFB vs. MFB, respectively (Figure 2a and Appendix A). The majority of the DEMs existing in MY vs. PR is consistent with the results of transcriptional DEGs (Figure 3a), suggesting the transition from the mycelium to the primordium is the most drastic. A Venn diagram showing the common and unique DEMs between the three comparison groups indicated most of the DEMs were unique, with only 11 (1.47%) metabolites in common (Figure 2b), while a heatmap of the DEMs clearly reflected these changes (Appendix A). The top 20 up- and downregulated DEMs in the three comparisons are shown in Appendix A.

#### 3.1.3. K-Means Clustering and KEGG Enrichment Analysis of DEMs

To obtain the overall dynamics of the DEMs, 748 co-expressed differential metabolites were clustered, and the results showed that the DEMs can be classified into four different groups (C1–C4) based on their accumulation patterns, corresponding to the peak levels of the metabolites in the four developmental stages (Figure 2c). These results indicate the MY, PR, YFB, and MFB stages of *M. sextelata* have different metabolic composition patterns.

KEGG enrichment analysis of the DEMs showed that all three comparisons were enriched in those pathways involving ABC transporters, amino acid metabolism, fatty acid biosynthesis, and metabolism (Figure 2d–f). ABC transporters are a group of membrane proteins that transport sugars, amino acids, and nucleotides and may be involved in important physiological processes during the morphogenesis and maturation of morels, such as nutrient accumulation, lipid homeostasis, and signal transduction. The pathways enriched in MY vs. PR included amino sugar and nucleotide sugar metabolism (*p* = 0.035) and arginine and proline metabolism (*p* = 0.044) (Figure 2d and Appendix A). These metabolic pathways suggest the mycelium stage absorbed abundant nutrients for primordium and ascocarp formation. Aminoacyl-tRNA biosynthesis and cyanoamino acid metabolism were significantly enriched in PR vs. YFB (*p* = 0.0054 and *p* = 0.0055) (Figure 2e and Appendix A), indicating rapid synthesis of proteins. YFB vs. MFB were significantly enriched in tyrosine metabolism (*p* = 0.0058), glutathione metabolism (*p* = 0.0142), methane metabolism (*p* = 0.0172), cysteine and methionine metabolism (*p* = 0.0216), aminoacyl-tRNA biosynthesis (*p* = 0.0216), and 2-oxocarboxylic acid metabolism (*p* = 0.0379) (Figure 2f and Appendix A). These diverse pathways of amino acid metabolism may be related to the specific flavor of mature ascocarps [1,40]. In summary, the expression pattern of KEGG functional pathways and metabolic dynamics is functionally correlated to morphological development and ascocarp maturation.

### 3.2. Transcriptomics Analysis during Morphogenesis and Maturation in M. sextelata

#### 3.2.1. Quality Assessment of Transcriptome Data

To understand the gene regulation basis of metabolic differences during development and maturation in *M. sextelata*, the transcriptomes of 12 samples (four representative developmental stages) were sequenced. A total of 82.34 Gb of clean data with Q20 ≥ 97.33% and Q30 ≥ 92.87% were yielded after the removal of adapters and low-quality sequences (Appendix A). The FPKM was used to measure the expression level of the transcripts or genes. Pearson’s correlation coefficient was used to evaluate biological repeatability, and the results indicated stable relevance (Appendix A). All the biological replicates showed high correlation (R^2^ > 0.75). PCoA and PCA analysis showed that the three biological repeats at each stage clustered well together, with clear separation at different stages, indicating the RNA-seq data varied at different developmental stages (Figure 3a and Appendix A). These results indicate the quantity and quality of transcriptomes are sufficient for further analysis.

#### 3.2.2. Identification and Enrichment of DEGs

Differentially expressed genes (DEGs) of the three comparative groups with a |Log2(foldchange)| ≥ 1 and a Padj < 0.05 were selected. MY vs. PR contained the majority of the DEGs (1826 upregulated and 2163 downregulated), followed by PR vs. YFB (1637 upregulated and 795 downregulated), and YFB vs. MFB (742 upregulated and 620 downregulated) (Figure 3b). An expression heatmap of the DEGs during the different developmental stages is shown in Figure 3d, which shows that the different developmental stages have different gene expression patterns. Only 353 (6.61%) common DEGs were identified from all of the comparison groups (Figure 3c), indicating they persistently fluctuated during morel development and maturation.

To further explore the potential functions of the DEGs, GO annotation and enrichment analyses were performed. Upregulated genes were significantly enriched in nucleotide metabolism and ribosome-related terms, and downregulated genes were significantly enriched in signal transduction, membrane transport, and response to environmental information processes in MY vs. PR (Figure 3e and Appendix A). Peptide metabolism, biosynthetic processes, and ribosome-related terms were significantly enriched in PR vs. YFB (Figure 3f and Appendix A). In contrast, energy metabolic processes (ATP metabolic process, NADH metabolic process, and oxidoreductase activity) and cytochrome complex assembly were significantly upregulated in YFB vs. MFB, while ascospore formation, cytoskeleton, and cell polarity were significantly downregulated (Figure 3g and Appendix A).

KEGG enrichment analysis was also performed to identify the metabolic pathways and functions of the DEGs (Appendix A). The DEGs were mainly enriched in the metabolic pathways, especially carbohydrate metabolism and amino acid metabolism (Appendix A). These results indicate these functional pathways are active and fluctuate during development and maturation, providing directions to elucidate the molecular mechanisms of morphological development and ascocarp maturation in *M. sextelata*.

To understand what genes might be important for the whole fruiting body development, these comparisons (MY vs. PR, MY vs. YFB and MY vs. MFB) were further analyzed. The Venn diagram showed that the majority of the DEMs (26.09%, 185/709) and DEGs (43.29%, 2821/6517) were shared among the three comparisons (Appendix A), and the 2821 shared DEGs were grouped into two clusters (Appendix A). Cluster 1 contained 1419 genes that were highly expressed in MY, while cluster 2 contained 1402 upregulated DEGs in the PR, YFB, and MFB stages, and might play key roles in the whole fruiting body development. Beyond providing statistical data for the DEGs and DEMs, we further scrutinized the biological function of the 1402 upregulated DEGs in the PR, YFB, and MFB stages. The top 20 enriched GO terms are shown in Appendix A, which can be summarized as being related to (1) metabolic processes containing carbohydrates, amino acids, small molecules, energy, etc.; (2) fungal cell wall components; and (3) response to stimuli. The above processes could be crucial for the whole fruiting body development of *M. sextelata*, so specific cases such as carbohydrate-active enzymes and transcription factors, cell wall synthesis and remodeling, and signal transduction and communication were focused on further in the following analysis.

#### 3.2.3. Analysis of Transcription Factors and CAZymes

Transcription factors (TFs) are key regulators that activate physiological and metabolic responses. In this study, a total of 254 transcription factors are detected and categorized into nineteen families (Appendix A), and the top five TF families by count are shown in Figure 4a. The C2H2 and Zn-clus dominated, including 95 (37.40%) and 77 (30.31%) transcripts, respectively. Similarly, among the 133 DEGs annotated as TFs, C2H2 and Zn-clus were also more abundant, containing 58 (44.36%) and 39 (29.32%) transcripts, respectively. The C2H2 domains belong to the superfamily of regulatory transcriptional factors that contain finger-like motifs and are involved in carbon catabolite repression, acetamide regulation, and differentiation of the fruiting body. Zn-clus encodes the DNA-binding domains associated with a number of fungal transcription factors.

The C2H2 and Zn-clus trends during the different developmental stages are shown in Figure 4b, which are classified into three clusters, reflecting high expression at the mycelium, primordium, and fruiting body stages, respectively. The proportions of C2H2 (48.78%) and Zn-clus (51.22%) were comparable among the highly expressed transcription factors in the mycelium stage; however, C2H2 (75.00%) was dominant among the highly expressed transcription factors in the primordium stage, and C2H2 (62.50%) was also dominant among the highly expressed transcription factors in the fruiting body stage. These results indicated C2H2 might be more involved in primordium formation and fruiting body maturation, and the transcription factors we identified were valuable candidate regulators of morphogenesis and maturation in *M. sextelata*. To validate the expression patterns of these TFs, we randomly selected six differentially expressed TFs for detection by qRT-PCR. The similar expression patterns of the selected TFs were consistent with their FRKM values in the transcriptomic data, confirming the accuracy of the transcriptomic data and repeatability of the expression patterns (Figure 4c). Details of the primers are shown in Appendix A.

Carbohydrates, also known as saccharides, are the main source of energy required by organisms to sustain life activities. CAZymes are enzymes that degrade, modify and create glycosidic bonds. In this study, a total of 372 CAZymes were annotated (Appendix A), of which glycoside hydrolases (GH) were dominant (44.62%), followed by auxiliary activities (AA; 20.16%), glycosyltransferases (GT; 18.55%), carbohydrate esterases (CE; 6.45%), polysaccharide lyases (PL; 5.65%), and carbohydrate-binding modules (CBM; 4.57%). There were 236 CAZymes identified as differentially expressed genes during the growth of *M. sextelata*, and GH was also more abundant, containing 108 differentially expressed genes (45.76%), indicating active hydrolysis and rearrangement of glycosidic bonds.

In addition, the CAZymes trends during the different developmental stages are shown in Appendix A, which are classified into three independent clusters, reflecting the high expression in the mycelium, primordium, and fruiting body stages, respectively. Moreover, highly expressed CAZymes were more abundant in the mycelium stage (45.34%), followed by the primordium (27.54%), and fruiting body (27.12%) stages. The results were also supported by the fact that the abundant exogenous nutrients were degraded and assimilated during the physiological process of the MY stage, while the primordium and fruiting body stages were mainly involved in the synthesis and modification of the endogenous components. In summary, these results indicate varied and diverse CAZymes are required for the morphological development and ascocarp maturation of *M. sextelata*.

### 3.3. Integrated Metabolomics and Transcriptomics Analysis

To understand the relationship between gene expression patterns and differences in metabolites, DEGs and DEMs with Pearson’s correlation coefficients (PCCs) > 0.95 were used to generate nine-quadrant plots (DEGs, |Log2(foldchange)| ≥ 2; DEMs, |Log2(foldchange)| ≥ 1). In the diagram, genes and metabolites with no difference were located in quadrant 5, genes and metabolites with a positive correlation were located in quadrants 3 and 7, and those with a negative correlation were located in quadrants 1 and 9. The results highlight significant correlations between the DEMs and DEGs (Figure 5a–c), indicating the metabolic dynamics may be directly or indirectly regulated by the corresponding DEGs.

In addition, KEGG co-enrichment analysis was performed based on the DEGs and DEMs. The results showed that both the DEMs and DEGs were significantly enriched in amino sugar and nucleotide sugar metabolism and arginine and proline metabolism in MY vs. PR (Figure 5d). In PR vs. YFB, the DEMs were significantly enriched in the aminoacyl-tRNA biosynthesis and cyanoamino acid metabolism pathways, and the DEGs were significantly enriched in the amino acid metabolism and nitrogen metabolism-related pathways (Figure 5e). In YFB vs. MFB, both the DEMs and DEGs were enriched in the glutathione metabolism and cysteine and methionine metabolism pathways (Figure 5f). The integrated analysis indicates pathways related to carbohydrate metabolism, amino acid metabolism, nucleotide metabolism, and lipid metabolism may play important roles in morphological development and ascocarp maturation in *M. sextelata*.

## 4. Discussion

Morels are highly popular edible mushrooms with great scientific and economic value [2]. Clarifying the physiological and molecular mechanisms behind the morphogenesis and maturation of morels will be helpful for breeding practices and artificial cultivation [9]. In this study, metabolomics and transcriptomics were used to construct molecular regulation profiles of MY, PR, YFB, and MFB in *M. sextelata*.

Metabolomic data identified a total of 943 metabolites, mainly including lipids and lipid-like molecules, organic acids, and derivatives (Figure 1b and Appendix A), which was consistent with the reported metabolomic studies of morels [41,42,43]. In addition, our study placed emphasis on the correlation between the metabolic and transcriptomic data and focused on the dynamic transition from vegetative to reproductive growth, which has not been touched on in previous studies [41,42,43]. Lipids have been recognized as a macronutrient, capable of energy storage [44], cell membrane formation [45], and membrane information transport [46]. It was also confirmed lipids were accumulated as an energy-rich substance in the sclerotia and hyphal cells of morels [47,48]. Organic acids have significant biological activities and are important components involved in organic synthesis and metabolism [49]. These identified metabolites provide insight into the chemical basis of the nutritional composition and specific flavor in morels. However, it is noteworthy that the detection limitations of the non-targeted methods and the database-based metabolite annotation also result in some biases (Appendix A). Therefore, further metabolomic studies are still needed to narrow down the key regulatory chemicals of morel development.

Transcriptomic data showed that the primordium differentiation stage was enriched in signal transduction, membrane transport, cellular communication, and response to environmental information processes (Figure 3e), indicating that the corresponding DEGs may be involved in primordium formation and differentiation, and that communication with environmental signals was more frequent during the PR stage. However, the YFB and MFB stages were enriched in the DEGs’ energy and amino acid metabolism in favor of the growth and maturation of ascocarps. Transcription factors (TFs) are key regulators that activate the physiological and metabolic responses of mushrooms [50,51]. Herein, we identified candidate transcription factors that may be critical for the growth and development of morels (Figure 4). C2H2 TFs regulate various developmental processes and abiotic stress responses, which are reported to be involved in fungal primordium and ascocarp formation [52]. Zn-clus TFs encode DNA-binding proteins associated with a number of fungal transcription factors [53]. These transcription factors may play important regulatory roles in the morphological development and ascocarp maturation of morels, and may help to elucidate the molecular mechanisms from vegetative to reproductive growth in the future.

Based on the dynamic metabolome and transcriptome profiles of the morphogenesis and maturation of *M. sextelata*, a conceptual model of the transcriptional and metabolic regulation of morels could be inferred for the first time (Figure 6a), which may provide insights into the molecular mechanisms of morphogenesis and maturation in the future.

In the model, environmental factors (e.g., temperature, humidity, nutrition, oxygen, and light) activate the regulation of transcription factors, which then promote the metabolic and transcriptional regulation of the morphogenesis and maturation processes. Specifically, the vegetative mycelium accumulates and stores abundant nutrients for fruiting; the transition from vegetative to reproductive growth involves a response to environmental information, and high expression of the signal transduction, membrane transport, and cell communication pathways may promote primordium formation; ascocarp maturation undergoes rapid morphological growth and accumulation of cytochrome, accompanied by spore formation for reproduction. Referring to the reported fungal studies on fruiting body development [54,55], genes related to cell wall biosynthesis and modification, as well as signal transduction and communication, were analyzed and discussed further, which have been shown to be important regulatory components in the complex morphogenesis of fungi. The expression patterns of these gene families in *M. sextelata* are shown in Figure 6b,c.

The results showed that the cell wall biosynthesis-related genes exhibited developmentally tissue-specific patterns (Figure 6b). The chitin biosynthetic enzyme families (represented by gene IDs 04835 and 10031) showed upregulated and high expression in the vegetative mycelium compared to the primordium and fruiting body, while the β-glucan biosynthesis families (represented by gene IDs 05406, 10243 and 00947) showed upregulation and high expression in the fruiting body relative to the vegetative mycelium. These results potentially indicated the cell wall composition was altered with the fruiting body development of *M. sextelata*. Genes related to cell wall remodeling were differentially expressed during mycelium, primordium, and fruiting body development, with the major chitin-activating enzyme families (61.54%) and glucan-linked families (65.22%) being upregulated during primordium and fruiting body development. Cross-linking or chemical modification of these enzyme families could influence the rigidity of the cell wall structure, which would be beneficial for maintaining the shape of the fruiting body. Overall, the regulatory interactions of these gene families might result in the formation of stage- and tissue-specific cell wall structures of morels, thus contributing to the completion of the complex multicellular shape of *M. sextelata*.

Response to signals from the environment or neighboring cells is essential for multicellular organisms to coordinate their development. Therefore, genes related to signal transduction and cellular communication were analyzed further in *M. sextelata*. The results showed that signal transduction and cellular communication-related genes also exhibited developmentally tissue-specific patterns, with 67 DEGs upregulated and highly expressed in the primordium and fruiting body (PR, YFB and MFB) compared to the mycelium (Figure 6c). Interestingly, 14 genes were highly expressed only in the primordium relative to the mycelium and fruiting body, involving transcription factors (represented by gene IDs 02650, 03504, 01604 and 10179), immune response (represented by gene IDs 04080, 01799, 02633 and 05830), protein kinases (represented by gene IDs 01392 and 00547), and membrane transporters (represented by gene IDs 10569, 06960, 07776 and 07651), and the synergistic regulation of these gene families was associated with developmental signaling and might have participated in the primordium initiation. Mating-type genes were also analyzed, which were involved in the fungal sexual reproduction and meiosis processes. The heatmap results showed that both MAT1-1-1 and MAT1-2-1 were highly expressed in the primordium stage, compared to the mycelium and fruiting body stages (Appendix A). Previous studies have shown that the mating-type genes of morels could influence the microstructure of the fruiting body [56], so highly expressed mating-type genes in the PR stage might promote the rapid differentiation to pileus and stipe, which was further involved in the fruiting body development and meiosis processes. However, it is still not enough to address how mating-type genes regulate fruiting body development, which needs to be further studied.

How fungi sense environmental signals to shift from nutritional to reproductive growth is a research focus in mycology [57]. Similar to common edible mushrooms, the nutritional and reproductive growth of morels show very different features and dynamics (Figure 1a and Figure 6). Primordium formation is considered to be a critical “intermediate” or “transitional” phase for successful differentiation into ascocarps [58,59,60]. Our results support that primordium formation is triggered by changing comprehensive environmental information (e.g., nutrient availability, temperature, light, humidity, and oxygen) and involves a transition from nutritional mycelia to primordia, which differentiate into reproductive complex structures. Ascocarp maturation is associated with the accumulation of metabolites, which leads to changes in appearance, texture, and flavor [61]. The young fruiting body is associated with rapid morphological growth, such as energy metabolism, carbohydrate synthesis, protein synthesis, and cytochrome assembly, whereas the mature fruiting body is associated with morphological stability and spore formation, such as cytoskeletal stability, maintenance of cell polarity, and ascospore formation. Interestingly, ascocarp color is one of the most important commercial traits and may account for cytochrome accumulation in morels [62]. In addition, the apparent phototropism may result from the high expression of cell polarity maintenance [63,64].

In conclusion, our study revealed the dynamics of the metabolome and transcriptome during morphogenesis and maturation in *M. sextelata*. A conceptual model of transcriptional and metabolic regulation was inferred during morphogenesis and maturation in *M. sextelata* for the first time. These datasets and results will contribute to unraveling the complex regulatory mechanisms from vegetative to reproductive growth, and the future identification of key signaling molecules and regulatory genes will be necessary to improve the fruiting stability and artificial cultivation of morels.

## Figures and Tables

**Figure 1 jof-09-01143-f001:**
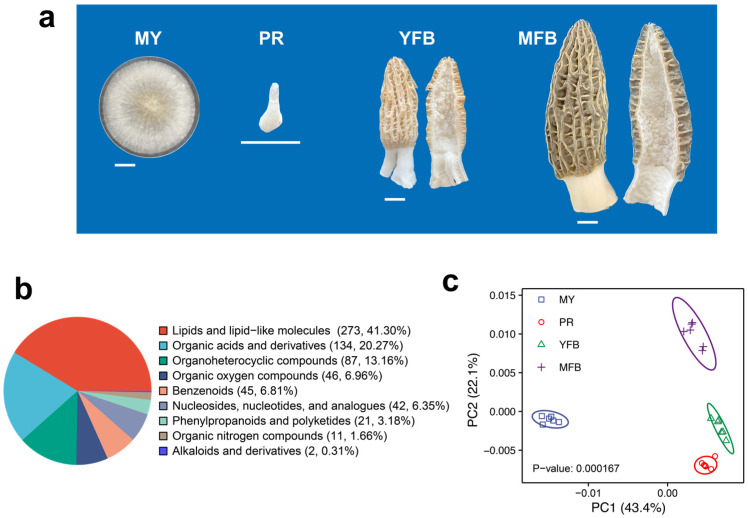
Morphological phenotypes and comparative classification of all metabolites during the development and maturation of *Morchella sextelata*. (**a**) Representative phenotype photos of MY, PR, YFB, and MFB. MY, mycelium stage; PR, primordium differentiation stage; YFB, young fruiting body stage; MFB, mature fruiting body stage. Size bars = 1 cm. (**b**) Classification of all metabolites. (**c**) PCA analysis of metabolites composition at different stages.

**Figure 2 jof-09-01143-f002:**
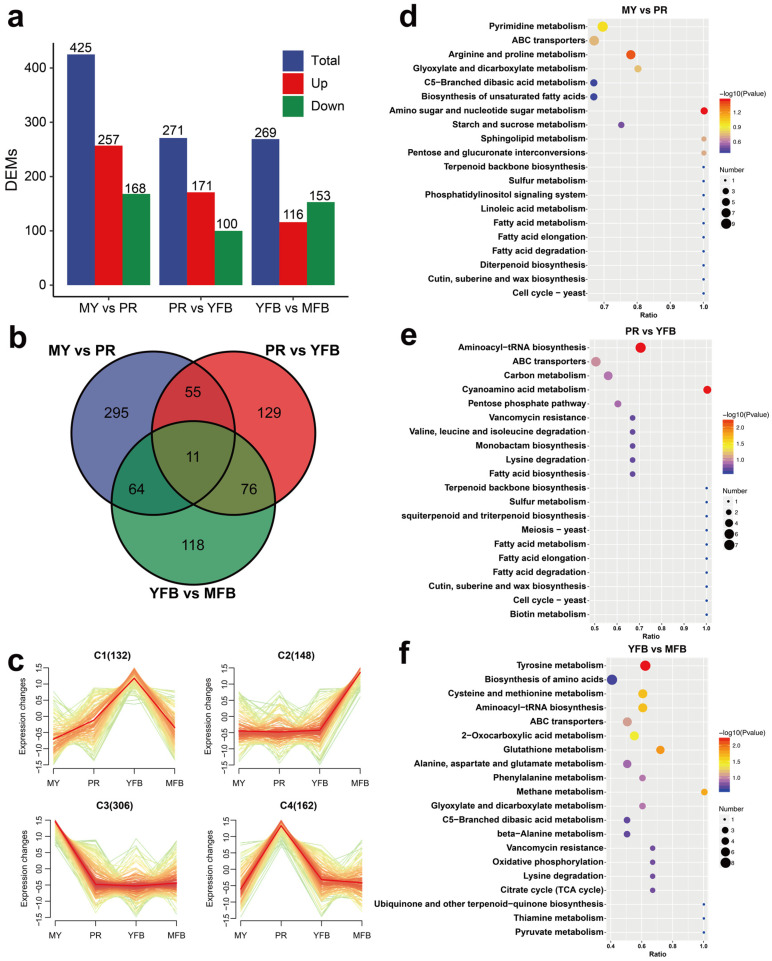
Metabolomics analysis during morphogenesis and maturation of *M. sextelata*. (**a**) Statistical analysis of DEMs. Blue represents the total DEMs, red represents the upregulated metabolites, and green represents the downregulated metabolites. (**b**) Venn diagram of the DEMs among the different developmental stages. (**c**) K-means cluster analysis of the DEM expression patterns. (**d**–**f**) KEGG enrichment analysis of the DEMs in MY vs. PR (**d**), PR vs. YFB (**e**), and YFB vs. MFB (**f**). The dot size represents the gene counts, and the color represents the *p*-value.

**Figure 3 jof-09-01143-f003:**
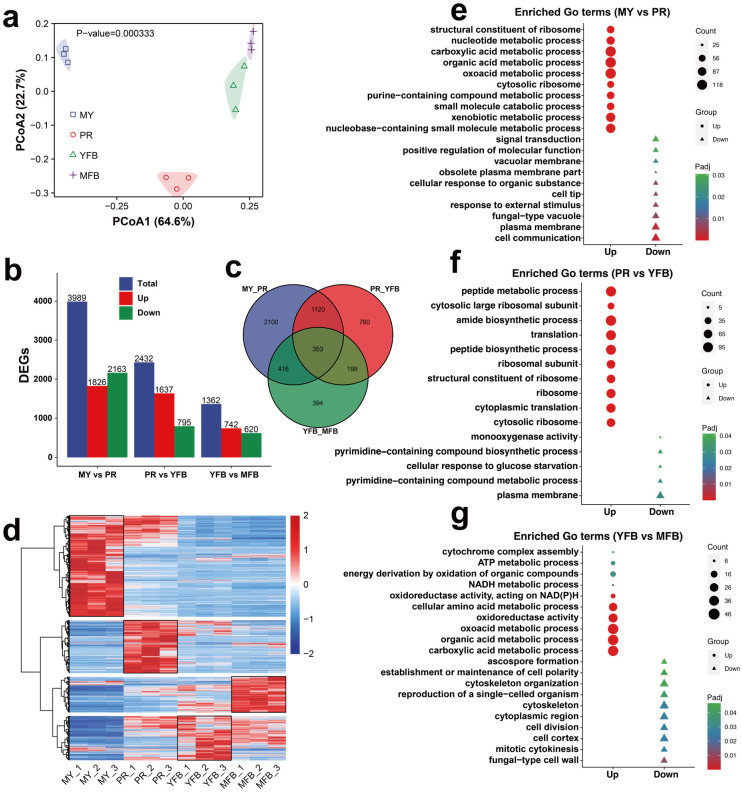
Transcriptomics analysis during morphogenesis and maturation of *M. sextelata*. (**a**) PCoA analysis of the gene expression profiles at different stages. (**b**) Statistical analysis of the DEGs. Blue represents the total DEGs, red represents the upregulated genes, and green represents the downregulated genes. (**c**) Venn diagram of the DEGs among the different developmental stages. (**d**) Heatmap showing the DEGs during the MY, PR, YFB, and MFB stages. Red represents the upregulated genes and blue represents the downregulated genes. (**e**–**g**) GO enrichment analysis of DEGs in MY vs. PR (**e**), PR vs. YFB (**f**), and YFB vs. MFB (**g**). Triangles represent upregulated genes and circles represent downregulated genes. The shape size represents the gene counts, and the color represents the Padj.

**Figure 4 jof-09-01143-f004:**
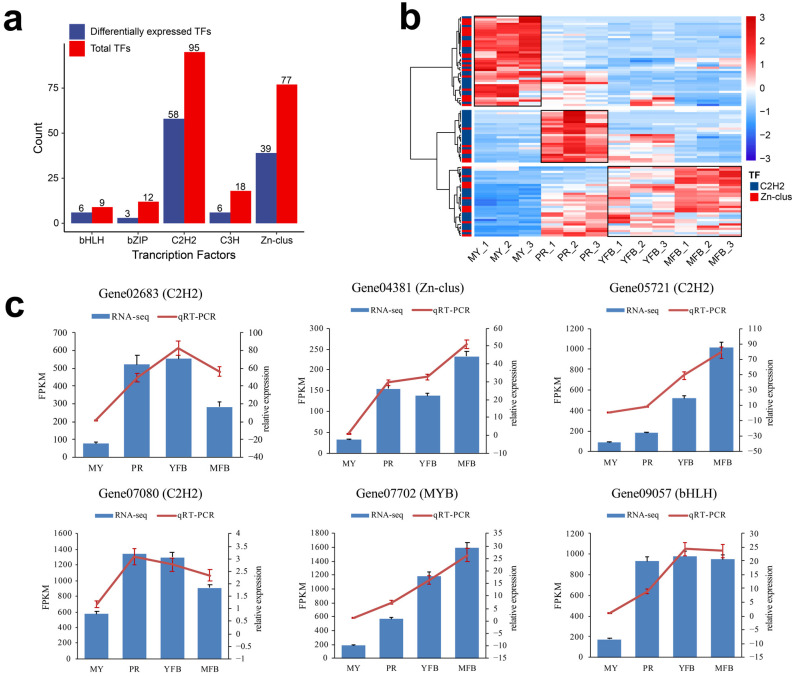
Analysis and qRT-PCR validation of the transcription factors. (**a**) The distribution of the top 5 TF families. Red represents the total TFs, and blue represents the differentially expressed TFs. (**b**) Heatmap showing differentially expressed C2H2 and Zn-clus. Red represents the upregulated genes, and blue represents the downregulated genes. (**c**) qRT-PCR validation of the expression patterns of the TFs that participated in the regulation of morphogenesis and maturation in *M. sextelata*. The bar graphs present the results of the RNA-seq, and the line graphs present the qRT-PCR results. The scale on the left axis represents the FPKM value, and the scale on the right axis represents the relative expression level. Data are the means ± SD of three biological replicates.

**Figure 5 jof-09-01143-f005:**
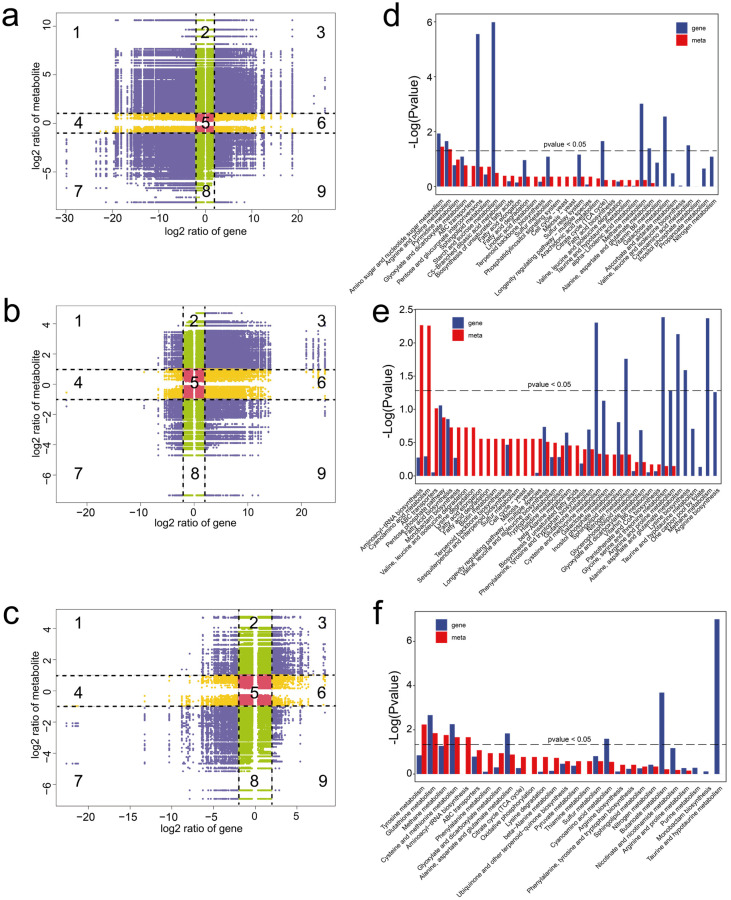
Integrated analysis of the metabolomes and transcriptomes of *M. sextelata*. (**a**–**c**) Nine-quadrant plots showing correlation analysis in MY vs. PR (**a**), PR vs. YFB (**b**), and YFB vs. MFB (**c**). Numbers (1 to 9) represent different quadrants. Purple represents significant changes in both the DEGs (|Log2(foldchange)| ≥ 2) and DEMs (|Log2(foldchange)| ≥ 1), green represents significant changes in the DEMs (|Log2(foldchange)| ≥ 1) and no significant changes in the DEGs (|Log2(foldchange)| < 2), yellow represents significant changes in the DEGs (|Log2(foldchange)| ≥ 2) and no significant changes in the DEMs (|Log2(foldchange)| < 1), and red represents no significant changes in either the DEGs (|Log2(foldchange)| < 2) or DEMs (|Log2(foldchange)| < 1). (**d**–**f**) KEGG enrichment analysis of the DEGs and DEMs were expressed in MY vs. PR (**d**), PR vs. YFB (**e**), and YFB vs. MFB (**f**). Gene represents the enrichment results of the DEGs, and meta represents the enrichment results of the DEMs.

**Figure 6 jof-09-01143-f006:**
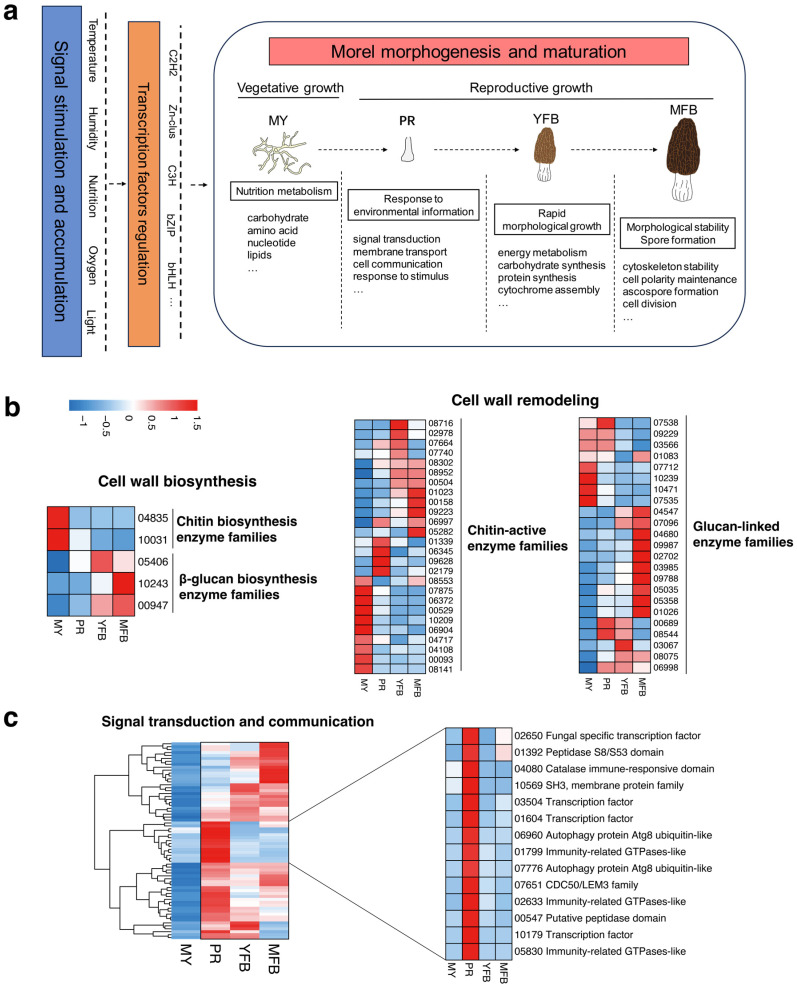
(**a**) A conceptual model of transcriptional and metabolic regulation during morphological development and ascocarp maturation in *M. sextelata*. Blue represents signal stimulation and accumulation of environmental factors (e.g., temperature, humidity, nutrition, oxygen and light); brown represents candidate transcription factor regulation (e.g., C2H2, Zn-clus, C3H, bZIP and bHLH). (**b**) Expression heatmap of the cell wall biosynthesis and remodeling-related genes in *M. sextelata*. (**c**) Expression heatmap of the signal transduction and communication-related genes in *M. sextelata*. Genes are denoted by gene IDs. Blue and red colors represent low and high expression, respectively. MY, mycelium stage; PR, primordium differentiation stage; YFB, young fruiting body stage; MFB, mature fruiting body stage.

## Data Availability

The RNA-seq data in this study have been deposited in the NCBI repository (https://www.ncbi.nlm.nih.gov/ (accessed on 29 June 2023)), accession number PRJNA989072.

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
