# Peer review of "Integration of Metabolomes and Transcriptomes Provides Insights into Morphogenesis and Maturation in Morchella sextelata"

_jof, 2023, doi:10.3390/jof9121143_

Round 1

Reviewer 1 Report

Comments and Suggestions for Authors

This work gaves an important contribution in the knowledge of transcriptional and metabolic regulation in the different phases of Morchella life cycle. The discussion focuses on these aspects and little attention is payed to most of the genes which are more directly related to the hyphal morphological changes such as  genes involved in cell wall remodeling and  adhesion. So I suggest the authors to change the introduction at line 72:

“To explore the molecular regulation mechanisms of morphological development and  ascocarp maturation in M. sextelata” in a sentence like: To explore the transcriptional and metabolic regulation in the different morphological phases of M. sextelata and during ascocarp maturation…..”

The article is well structured and clearly written. I only suggest at line 73 to define the abbreviation at this first mention

Author Response

Thank you very much for your insightful and helpful comments. We have revised the manuscript according to your comments and suggestions. Please see the attachment for point-by-point responses.

Reviewer 2 Report

Comments and Suggestions for Authors

The manuscript titled “Integration of metabolome and transcriptome provides insights into morphogenesis and maturation in Morchella sextelata” presents a classical developmental biology study using metabolomics and transcriptomics. Morel cultivation has a growing market due to its culinary and conceivably medicinal properties therefore the subject of the article is relevant and addresses an important question: what is the genetic program of the morel fruiting body development? The authors performed transcriptomics and metabolomics on four developmental stages (Vegetative mycelium - VM, Primordium - PR, Young Fruiting Body - YFB and Mature Fruiting Body - MFB). They calculated differential expressed genes (DEGs) and metabolites (DEMs) along the developmental stages. The authors also performed GO enrichment analysis within the DEGs and DEMs.

While the methodology is acceptable (but see my comments below), the results are too broad and I don’t think that the findings provide any additional information that leads us closer to understanding the fruiting development in morels. This can be seen well in Figure 6 where the findings of the study are summarised. I have to say that the idea of the figure is excellent I really like it, but the content is not met with the Journal’s scientific standards. For example, we have known for decades that fruiting body development is triggered by environmental stimulations and that transcription factors are needed to change expression profiles therefore the provided information is not new to the science. Moreover, the morel morphogenesis and maturation part describes faulty information. For example, in the primordium stage signal transduction is listed as a possible process, but signal transduction is down-regulated in this stage relative to the vegetative mycelium! Unfortunately, the study does not go beyond the GO enrichment analyses, therefore, it is unable to deliver detailed information about fruiting body development. I think reanalysing the data and performing a few additional analyses could provide substantial improvement thus giving better and more detailed insights about the genetics of fruiting body development in morels. My suggestions are the following:

1) Instead of only comparing subsequent stages authors could focus on two aspects of development: fruiting body initiation (MY vs PR) and fruiting body development (MY vs PR or YFB or MFB). The questions in these comparisons could be: what genes are upregulated in PR relative to the MY?  Are these genes still upregulated in subsequent stages (YFB, MFB) relative to the MY? These genes could be important throughout the whole development. Genes upregulated only in PR could be just triggers that initiate the whole development.

2) I encourage the authors to group genes based on InterPro domains because it could give more detailed information than GO. For example, CAZyms could be analysed separately to show fruiting body development-specific CAZyms that have already been described among Agaricomycetes. A particular focus could be receptors: is there any sign that receptors are upregulated at the PR stage? If yes could the authors further specify what receptors are playing a significant role in the primordium development? This would be really exciting and the left side of Figure 6 could be updated with more specific signal transduction processes. Mitosis and meiosis-related genes could be analysed separately too: is the total expression of these genes higher during the fruiting body development than in the mycelium? What about the mating type genes? Do they find similar patterns to Liu et al 2022 (https://www.mdpi.com/2309-608X/8/6/564).

3) It would be nice to integrate more of the knowledge that has been piled up in the past years on the development biology of morels and other fungal taxa. One way could be if the authors included existing data in their analyses. For example, the transcriptomic data of Deng et al. 2022 (https://www.frontiersin.org/articles/10.3389/fgene.2021.829379/full) could be combined by the authors’ data increasing the amount of available transcriptome from mature stages. Note that in Deng et al.’s data, the PMP_3 seems to be an outlier sample, therefore, when the authors reanalyse it, and find it to be an outlier too, they should exclude this from their analysis.

Overall, the authors could go beyond the KEGG and GO analysis and comparing the results of other fruiting body development studies would increase the value of the manuscript. I suggest incorporating the following articles into the discussion, along with central questions about fruiting body development. I suggest finding orthologous genes of other species that are essential components of fruiting body development and examining the expression pattern of these genes in M. sextelata. Please find fruiting body development-related genes in the following articles, among others:

Lütkenhaus et al. 2019. Genetics. https://pubmed.ncbi.nlm.nih.gov/31604798/

Nagy et al. 2023. Studies in Mycology. https://www.ingentaconnect.com/content/wfbi/sim/2023/00000104/00000001/art00002;jsessionid=2ghqkgvqof1j7.x-ic-live-02#

Krizsán et al. 2019. PNAS. https://www.pnas.org/doi/abs/10.1073/pnas.1817822116

Short comments:

Enrichment analyses should be done on up- and down-regulated metabolites separately as it was performed in the case of transcriptomics.

In general, instead of the “developmental period”, I would use the “developmental stage”. In developmental biology, “stage” is used to describe a temporal phase of a morphological structure.

Generally, in written English, the numbers below 10 should be written out in words. Please correct this in the manuscript if the journal does not have an opposite instruction.

I found that the methodology of the transcriptomic analyses was a bit vague. For example, what normalisation factor did they use in the DESeq analysis? Did they use raw reads as input in the DESeq analysis? Currently, the methods suggest that they used FPKM values in DESeq2, is that true? What kind of dispersion estimation method was applied? Did the authors perform any prefiltering on the raw read counts? If not, it is always a good idea to filter low-counted reads. Did they use any shrinkage estimator to calculate fold change? Did they perform independent hypothesis weighting?

Line 19. Place a decimal separator in the number 5342: 5,342

Lines 39-41. Could you provide a number in tonnes or a value of how big is the morel market? And then contrast it with other commercial mushrooms? It would be nice to place its economic importance like this.

Line 44. 3 evolutionary clades à three evolutionary clades

Line 137. What is VIP?

Line 139. Provide the version number of the package

Line 139-140. How did you perform an enrichment analysis? Using Fisher test or Hypergeometric test or something else? Please describe it.

Line 154. What was the quality threshold or procedure for quality filtering?

Line 155. What genome did the authors use to map the reads? Provide clear reference to the genome.

161-163. Can you please provide some more information about how you identified TFs. Based on sequence similarity, HMM search or something else?

Lines 170-171. “cor program from R”. cor is a function in R. So I would phrase this something like this “The correlation between metabolome and transcriptome was assessed by the cor function of the stats v. [version number] R package by applying a 0.95 PCC threshold.”

Lines 200-201: What do you call a significant separation? Did you perform a statistical test, if yes please report the results of it.

Line 212. Venn diagram instead of Venn plot

Line 228-230. I don’t see what results would lead to this conclusion. First of all, isn’t ascocarp maturation a morphological development, too? Second, what is significantly different? The amount of metabolite, composition stb..? Third, If something is significant, I would expect a statistical test.

Line 231. Delete “groups”. I think “all three comparisons were enriched in pathways..” would make more sense.

Lines 258-259.  “stable and convincing reproducibility” please report an exact number such as “All the biological replicates showed high correlation (R2> 0.8)” Also I have to mention that the YFB_1 sample is more similar to MFB_1 than the other two YFB stages.

Line 303. What are “Top5 TFs families”? Top 5 biggest family? Please elaborate on this.

Line 311. “into 3” change to “into three”

Lines 310-313. This sentence is inconclusive: if DEGs are depicted, I would expect high expression in each of the stages. Rather than the original conclusion, I would explore if there is a specific type of TF that is more representative of any of the stages. You might perform an enrichment analysis to show this. For example, PR stage seems to be enriched in C2H2 TFs.

Lines 333-335: What are the different quadrants? It is not clear. It might be clarified on the Fig 5.

Line 549. It seems that there is a mistake in the author’s name in this citation

Figure 5a-c: what do the different colours represent?

Comments on the Quality of English Language

The English is good; I found only small mistakes indicated in my review.

Author Response

Thank you very much for your insightful and helpful comments. We have carefully revised the manuscript according to your comments and suggestions. Please see the attachment for point-by-point responses.

Round 2

Reviewer 2 Report

Comments and Suggestions for Authors

I would like to thank the authors for their detailed responses and for addressing most of the concerns I noted in the first version of the manuscript. The second version has been improved substantially.

Please check more carefully the DESeq2 methods. Some statistical methods (Wald test vs shrinkage estimator) are not used in the right context. The method description still needs to be improved a bit. See my specific comments below.

The English could be revised by a native speaker.

Specific comments:

Line 161. The Wald test is for testing differential gene expression. The shrinkage estimator is to adjust log fold change values based on read counts. Three shrinkage estimators are implemented in DESeq2: normal, ashr, apeglm. Please specify which one you used.

Lines 163-164: The sentence needs to be grammatically corrected: “Genes … were identified as significantly differential expression.” Suggestion: “Genes … were identified as DEGs.”

Line 255-256: Is there any literature data about the metabolites of the flavour and scent of morels? If yes, could you link any of that to your data?

Line 346: Correct “6” to “six”

Line 480-481: “… genes exhibited developmentally tissue-specific” This is grammatically incorrect and probably a word is missing at the end. “tissue-specific patterns”?

Comments on the Quality of English Language

The English could be revised by a native speaker.

Author Response

Thank you very much for your helpful comments. We have carefully revised the manuscript according to your comments, and the manuscript has been polished by MDPI language editing services. Please see the attachment for point-by-point responses.
